# No evidence of reduced capacity during highly demanding cognitive tasks in healthy older adults at electroencephalographic risk of cognitive impairment

Jorge Sigg-Alonso[1], Mauricio González-López[1], Eduardo Gonzalez-Moreira[2], Rodolfo Solís-Vivanco[3,4], Thalía Fernández[1]*

1 Laboratory of Psychophysiology, Department of Behavioral and Cognitive Neurobiology, Instituto de Neurobiología, Universidad Nacional Autónoma de México (UNAM) Campus Juriquilla, Querétaro, México, 2 Center for Biomedical Imaging and Neuromodulation, Nathan Kline Institute for Psychiatric Research, Orangeburg, New York, United States of America, 3 Laboratory of Cognitive and Clinical Neurophysiology, Instituto Nacional de Neurología y Neurocirugía Manuel Velasco Suárez, Ciudad de México, México, 4 Faculty of Psychology, Universidad Nacional Autónoma de México, Ciudad de México, México

* thaliafh@yahoo.com.mx

## Abstract

Healthy older adults with excessive theta absolute power (AP) are considered at electroencephalographic risk of developing cognitive impairment 7 to 10 years later. Although this population may exhibit a normotypic cognitive state, as revealed by traditional neuropsychological assessment, less is known about their performance during tasks with high cognitive demand and whether the degree of excessive theta AP can be used to predict their performance, which was our objective. We compared the scores from highly demanding memory tasks (i.e., the Visual Short-Term Memory Binding Test (VSTMBT) and the Loewenstein-Acevedo Scale for Semantic Interference and Learning (LASSI-II)) between older adults with and without excessive theta AP. No significant differences were found between the groups for any test score or for the predictive value of the theta AP for performance. The results of this study provide evidence that older adults with excessive theta APs do not exhibit impaired performance in highly demanding cognitive contexts. The possible role of cognitive reserve in alleviating evidence of deterioration is discussed.

## Introduction

As a result of population growth and aging, the World Health Organization estimates that, by 2030, the number of adults over 60 will reach 65.7 million people worldwide [1]. In Mexico, it is estimated that this number will reach 20 million [2]. Advanced age is the main risk factor for the development of neurocognitive disorders [1], which in severe cases result in disability and dependence, causing heavy emotional and economic costs at the individual, family, and social levels [3]. There is an increasing

**Data availability statement:** All data are available on the following link: https://github.com/JorgeSigg/No-evidence-of-reduced-capacity-during-highly-demanding-cognitive-tasks-in-older-adults-at-EEG-risk. The data base for the LASSI test is blocked and will be released once Roa's et al. (in preparation) paper on the standarization and normalization of the LASSI for the mexican population has been published.

**Funding:** This work was supported by grants IN225414, IN200817, and IT201123 from the Universidad Nacional Autónoma de México (DGAPA-PAPIIT), with TF as recipient. During this study, Jorge A. Sigg-Alonso (CVU 765024) was a beneficiary of a scholarship by Consejo Nacional de Humanidades, Ciencia y Tecnología (CONAHCYT). There was no additional external funding received for this study.

**Competing interests:** The authors have declared that no competing interests exist.

need to identify biomarkers capable of detecting the first physiological correlates of pathological cognitive aging before the onset of a neurocognitive disorder. Currently known biomarkers with the capacity to detect subtle changes include the concentrations of amyloid beta (Aβ42) and tau proteins in cerebrospinal fluid (CSF) and amyloid positron emission tomography features [4]. These biomarkers are invasive and expensive; therefore, there is a need to identify noninvasive and cost-effective measures sensitive to early subtle brain changes that occur in the preclinical phase of a neurocognitive disorder.

From a neurophysiological perspective, the resting-state electroencephalogram (EEG), which reflects the functional integrity of the brain, has been established as a sensitive neuroimaging technique to detect incipient changes in neural activity in the early stages of a neurocognitive disorder [5,6]. Prichep et al. [7] reported that healthy older adults with subjective memory complaints (Global Deterioration Scale = 2) and excessive absolute power (AP) in the theta frequency band had an increased risk of developing major cognitive impairment 7 to 10 years later. In addition, excessive theta power in older adults has been associated with increased Aβ deposition in the brain [8], decreased concentrations in CSF [9], decreased brain and hippocampal volumes [10], and progressive subjective cognitive decline [11]. Our research group studied this electroencephalographic risk marker in healthy older adults and identified EEG changes in both functional connectivity at rest [12] and event-related potentials during cognitive tasks [13,14]. In those studies, we did not find behavioral differences associated with high theta power, probably due to the use of traditional neuropsychological tests. However, Prichep et al. [7] reported that older adults who later exhibited cognitive deterioration and had higher theta APs also performed worse on attention tests, i.e., the Digit Span Test (forward and backward) and the Digit Symbol Substitution Test.

In addition to these neurophysiological approaches, efforts have been made to develop new neuropsychological tests that are capable of detecting subtle changes in cognitive processes during the preclinical phases of neurocognitive disorders [15–19], that is, when individuals report subjective cognitive decline that is not psychometrically objective and therefore does not meet the criteria for a neurocognitive diagnosis [20]. This effort has arisen because traditional neuropsychological tests have been shown to lack sensitivity for detecting subtle cognitive decline [19,21,22]. Recent evidence suggests that tasks with high cognitive demand, due to either interference control requirements or working memory load, may be sensitive to early cognitive changes in preclinical Alzheimer's disease [18,21]. Specifically, Parra et al. [21] reported that asymptomatic carriers of the E280A0 mutation of the presenilin 1 gene (familial Alzheimer's disease) performed worse than healthy adult controls did in the Visual Short-Term Memory Binding Test (VSTMBT), a visual working memory test with varying levels of perceptual load demand. Koppara et al. [22] also reported worse performance in the feature binding task (short-term memory binding condition) of the VSTMBT in subjects with cognitive complaints. In both studies, no differences were found between groups in other neuropsychological tests.

Similarly, Loewenstein et al. [18] reported significantly lower performance in older adults with subjective cognitive complaints (premild cognitive impairment [PreMCI]) than in controls using the Loewenstein-Acevedo Scale for Semantic Interference (LASSI), a verbal memory test requiring effective proactive and retroactive interference control. Additionally, Crocco et al. [23] reported that the percentage of semantic intrusions in this test distinguished subjects with PreMCI from controls. Loewenstein et al. [18] reported an association between increased Aβ load on positron emission tomography (PET) brain scans and failure to recover from proactive semantic interference as measured with the LASSI in otherwise neuropsychologically normal older adults. Using this test, they also reported associations among brain volume loss [24], hippocampal and precuneus cortical thinning [25], and failure to recover from proactive semantic interference in participants with early amnestic MCI.

Our study breaks new ground in this field by investigating the potential relationship between EEG-based biomarkers of cognitive impairment risk and neuropsychological tests that have demonstrated high sensitivity to the preclinical phases of neurocognitive disorders. Considering the predictive value of excessive theta activity for cognitive decline, the aim of this study was to fill a significant gap in the current literature. Our research goal was to determine whether there are discernible alterations in scores on highly demanding cognitive tests (i.e., the VSTMBT and LASSI-II) in healthy older adults with electroencephalographic risk of cognitive impairment (excessive theta AP) and whether the degree of theta AP predicts their performance.

## Materials and methods

The present study was approved by the Bioethics Committee of the Institute of Neurobiology (INB) at Universidad Nacional Autónoma de México (UNAM, REF 030-H-RM) in accordance with the guidelines established by the Declaration of Helsinki.

### Participants

The sample included 63 healthy adults over 55 years of age with no history of neurological disease or current psychiatric diseases. The inclusion criteria were as follows: a) at least 9 years of formal education; b) normal global functioning (Global Deterioration Scale score ≤2; [26]); c) an age-adjusted normal cognitive state (Montreal Cognitive Assessment (MoCA) score > 23; [27–29]); d) average intelligence (Wechsler Intelligence Scale for Adults Version IV score > 80; [30]); e) normal daily functioning (Instrumental Activities of Daily Living Scale score < 8; [31,32]); and f) normal levels of hemoglobin, cholesterol, triglycerides, glucose, and thyroid-stimulating hormone. All participants were screened for psychopathology with the 5.0.0 Spanish version of the MINI International Neuropsychiatric Interview [33] and a semistructured clinical interview performed by a clinical psychologist and a neuropsychologist. Although some participants scored 1 standard deviation below their normative group on some neuropsychological tests, none of them met Jak and Bondi's criterion for a mild neurocognitive disorder of > 1 SD below the norm in two tests per cognitive domain [34]. Participants were recruited from the general population through social media advertisements, radio, and recommendations from other participants. Participants were recruited from the 13th of April 2022 until the 4th of March 2024. All participants signed a letter of informed consent before starting any procedure.

### Neuropsychological instruments

A neuropsychological battery of traditional tests was designed to explore memory (Rey's complex figure [RCF], logical memory, and verbal paired associates [PAs]), executive functioning (phonological and semantic verbal fluency, mazes, Tower of Hanoi [3- and 4- disc versions], Trail Making Test [TMT] A and B, and Stroop), and intellectual capacity (WAIS-IV; [30]). Tests were selected from the Neuropsi Attention and Memory battery (NAM-2; [35]), the Neuropsychological Battery of Executive Functions and Frontal Lobes (BANFE- 2; [36]), and the versions of the TMT by Arango-Lasprilla et al. [37]

and Stroop de Rivera et al. [38]. In addition, the VSTMBT [39] and the Loewenstein-Acevedo Scale for Semantic Inter-ference and Learning II (LASSI-II; [18]; Roa et al., in preparation) were administered. The VSTMBT consists of a visual short-term memory task with two conditions: one requires the individual to remember the shape (single condition) of geometric stimuli, and the other requires them to remember both the color and the shape (binding condition). Both visual versions had a low cognitive load (remember two forms) and a high cognitive load (remember three forms) and were presented interleavedly (Fig 1). The percentage of correct responses was used for the analysis. After a warning stimulus (+), the figure to be remembered was presented for 2 s (encoding phase). A blank screen was presented for 1 s (retention phase), and finally, a second figure (probe) was presented. The participant reported verbally whether the encoding and probe figures matched. As soon as the participant gave an answer, the researcher presented the next trial. A total of 32 trials were performed for each condition.

The LASSI-II is a word learning test that requires individuals to learn 2 lists (i.e., A and B) of 15 words each, which contain three semantic categories: 5 fruits, 5 clothes, and 5 musical instruments. This test involves free recall (free recall A and B) and cued recall (cued recall A and B) after the first presentation of each list, a second learning trial for each list followed by a cued recall (cued recall A2 and cued recall B2), a free and cued recall of list A (free and cued recall A3) after cued recall B2 and a delayed recall of both lists. This test enables the assessment of maximum storage capacity (cued recall A2), the effects of proactive semantic interference (PSI) (cued recall B) and retroactive semantic interference (RSI) (cued recall A3), and recovery from PSI (r-PSI) (cued recall B2) after a second learning trial. The percentage of seman-tic intrusions was calculated for PSI and r-PSI, two measures that have been shown to be highly sensitive to cognitive decline. The percentage of PSI (% of intrusion errors on cued recall B) was calculated by dividing the total intrusions in cued recall B by the sum of total intrusions in cued recall B and total correct responses in cued recall B, and the percent-age of r-PSI (% of intrusion errors on cued recall B2) was calculated by dividing the total intrusions in cued recall B2 by the sum of total intrusions in cued recall B2 and total correct responses in cued recall B2 (as in [23]). The total number of intrusions was calculated as the sum of all intrusions in the complete test. The tests were administered to all participants in the same order by a neuropsychologist.

The full neuropsychological battery was administered in the morning, and standardized scores were calculated for each measure using the corresponding norms for the Mexican population (except for the VSTMBT and LASSI-II, which do not yet have normative data).

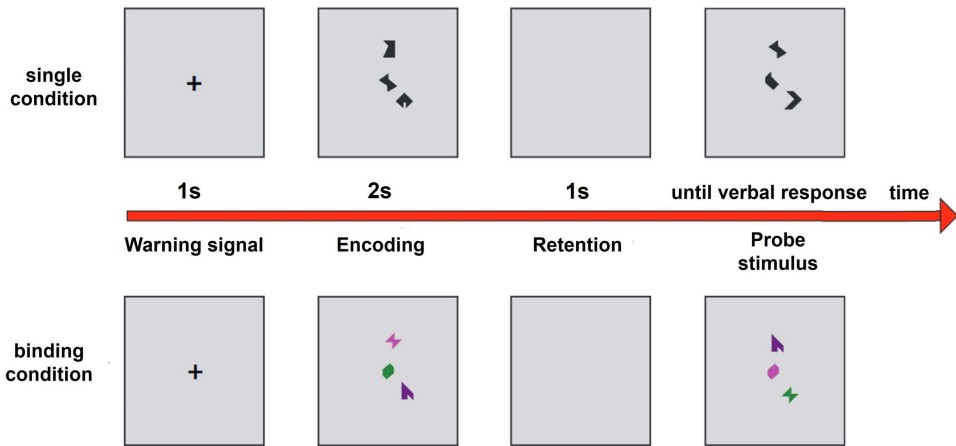

**Fig 1. Example of two VSTMBT items with high cognitive load.** The upper row shows a single-condition trial (shape). The lower row shows a binding-condition trial (shape and color). "+" is the fixation visual point. Adapted from [40].

### EEG recording and analysis

A resting 19-channel EEG was recorded via the Medicid IV system (Neuronic Mexicana, SA, Mexico) and the Track Walker TM v2.0 data capture system. The electrodes were mounted on an elastic cap (Electro-Cap, International, Inc., Eaton, Ohio, USA) according to the International 10–20 System and referenced to linked earlobes. Recordings were carried out in a sound-muffled, air-conditioned, and faradized chamber with a comfortable chair and dim lighting. The participants were asked to keep their eyes closed, relax (without goal-directed purposeful thinking) and not fall asleep. The EEG acquisition lasted 10 minutes, with a sampling rate of 200 Hz, a gain of 20,000, and impedances below 10 kΩ.

The EEG was visually inspected with Track Walker TM v2.0 software by two electroencephalography experts, and 24 epochs of 2.56 seconds were selected manually. These epochs were free of physiological and nonphysiological artifacts, as well as paroxysmal activity, and, in most participants, exhibited frequency and amplitude features of the posterior dominant alpha rhythm to avoid slow activity associated with drowsiness. Using Neuronic Quantitative Tomographic EEG software (Neuronic Mexicana S.A de C.V., version N_I_SW-5 6.2.4.0), the power spectrum was calculated from the selected epochs for each participant via fast Fourier transform with a spectral resolution of 0.39 Hz for each electrode and frequency bin (0.78–19.14 Hz). The spectral matrices were corrected by the global scale factor [41] to eliminate intrinsic nonneurophysiological interindividual differences. The AP was subsequently calculated at each electrode for the following frequency bands: delta [1.5–3.5 Hz], theta (3.5–7.5 Hz), alpha (7.5–12.5 Hz), and beta (12.5–19.5). Z values for AP were subsequently obtained according to values from a normative database [42]. The participants were classified into two groups: i) the control group (n = 27), which consisted of participants with a normal EEG, i.e., without abnormal waveforms and a normal quantitative EEG in all frequency bands ($-1.96 < z < 1.96$), and ii) the risk group (n = 36), which included participants with excessive theta AP ($z > 1.96$) in at least one electrode. Participants who exhibited any other abnormalities were excluded from the analysis.

To further study the relationship between excessive theta AP and cognition, the risk group was separated into two groups: the low-risk group (n = 18) and the high-risk group (n = 18). These groups were created by a median split according to four different criteria: the sum of the theta AP z value across all electrodes, the squared sum of the AP z value (maintaining the sign) across all electrodes, the mean of the theta AP z value across electrodes, and the number of electrodes with a theta AP z value $> 1.96$.

### Statistical analysis

A custom script for MATLAB R2020b (MathWorks, Natick, MA) was used for permutation statistical testing. The software Jasp 0.19.0 (https://jasp-stats.org) was used for the analysis of sociodemographic data, analysis of variance (ANOVA), principal component analysis (PCA), and regression analysis. All *p* values were corrected for multiple comparisons with the false discovery rate (FDR).

**Risk group vs. control group.** Comparisons of cognitive measures and normalized AP values for each frequency band between groups were performed via the multivariate permutation method (5,000 permutations). This nonparametric technique does not assume an *a priori* theoretical distribution since the distribution of null hypotheses of the statistical tests is generated iteratively through the random processing of the data. The probability values were corrected for multiple comparisons via the FDR. One subject was excluded from the group difference analyses involving APs because of data corruption in the EEG file.

For the analysis of continuous sociodemographic variables (age and years of schooling), the t test for independent samples was used. For the analysis of categorical variables (sex), the chi-square test ($x^2$) was used.

**High-risk group vs. low-risk group vs. control group.** One-way ANOVA was used for the group comparisons between the control and low- and high-risk groups in the highly demanding cognitive tests.

**Multiple regression analysis between theta and cognition.** Before the multiple regression analysis, a PCA of the theta AP z value of each electrode, using the varimax rotation method and a loading factor threshold of 0.7, was performed to identify the principal components (PCs) for use as predictors in the subsequent multiple regression analysis; the dependent variable was one cognitive score in each model. This analysis was performed to study the ability of excessive theta AP to predict performance on highly demanding memory tests and traditional neuropsychological tests.

## Results

### Demographic data

There were no significant differences between groups for age (risk: x̄ = 65.55 ± 7.45), control: x̄ = 65.03 ± 6.95), t = 0.28, $p = 0.97$), education (risk: x̄ = 19.26 ± 3.79), control: x̄ = 20.37 ± 4.36), t = -1.07, $p = 0.97$), or sex (risk female: 74.1%, control female: 69.4%; $x^2 = 0.16$, $p = 0.97$). In both groups, some individuals had scores below what was expected for their age on one of the traditional neuropsychological tests (control group: 44%, risk group: 47%), but they did not meet the criteria for a neurocognitive disorder [34].

### EEG data

Our group formation criterion was based on theta AP according to age-adjusted norms. Therefore, there were statistically significant differences between the risk and control groups in the mean z theta AP at all electrodes (all $p < 0.001$; x̄ = 7.33 ± 2.76). Within the risk group, the electrodes with significantly greater theta zAPs (z >1.96) were F8 (t = -5.33, $p < 0.001$; $d =$), F4 (t = -4.99, $p < 0.001$), and P3 (t = 4.93, $p < 0.001$; $d =$), whereas the control group had mean theta Z values close to 0 (Fig 2). There were no significant differences in the delta, alpha, or beta frequency bands.

### Cognitive results

**Risk group vs. control group.** We did not find significant differences between the groups in any traditional neuropsychological test, including memory and executive function (Table 1) or intelligence tests (supporting information S1 Table).

Similarly, we did not find significant differences between groups in any score of the VSTMBT (Table 2) or the LASSI-II test (Table 3), the total number of intrusions, the percentage of semantic intrusions (% of intrusion errors on cued recalls B and B2), or the recovery from interference (cued recall B2).

**High-risk group vs. low-risk group vs. control group.** When three groups were considered (high risk, low risk, and control), for any of the four separation criteria used, the one-way ANOVA used to compare the groups did not yield

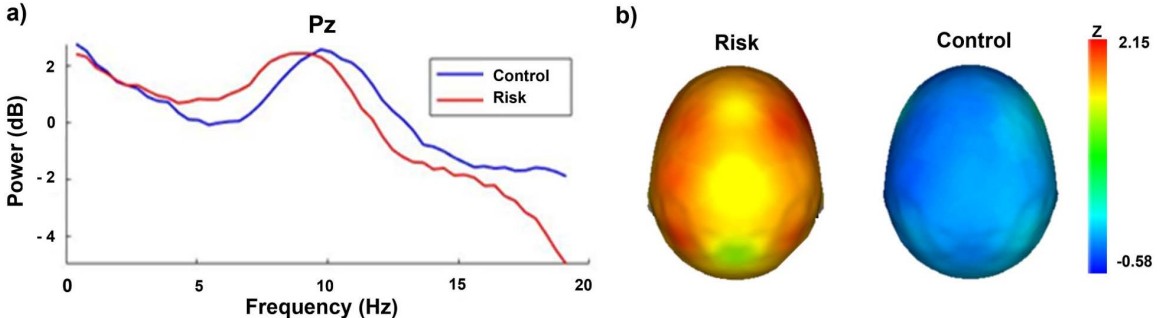

**Fig 2. Z-based theta (3.5–7.5 Hz) APs in both groups. a)** Power spectra for the risk (red) and control (blue) groups (Pz as a representative electrode). **b)** Mean z theta absolute power topographic maps**.**

**Table 1. Traditional neuropsychological tests.**

| | | Risk | Control | t value | *p*-FDR |
|---|---|---|---|---|---|
| | | Mean(SD) | Mean(SD) | | |
| Memory | RCF | | | | |
| | Copy | 10.86(1.80) | 9.77(2.10) | -2.14 | 0.65 |
| | Retrieval | 10.50(3.05) | 9.92(2.09) | -0.84 | 0.96 |
| | Logical memory | | | | |
| | Encoding | 13.77(2.84) | 13.48(2.24) | -0.43 | 0.96 |
| | Retrieval | 11.91(2.63) | 12.37(2.20) | -0.42 | 0.96 |
| | PA | | | | |
| | Encoding | 11.16(2.62) | 11.96(2.31) | 1.21 | 0.89 |
| | Retrieval | 13.33(2.90) | 13.03(2.20) | 0.71 | 0.96 |
| Executive | Semantic fluency | 12.66(3.52) | 12.51(3.09) | -0.19 | 0.97 |
| | Phonologic fluency | 14.05(3.25) | 13.29(2.99) | -0.92 | 0.96 |
| | Maze planning | 11.30(2.14) | 11.22(1.45) | -0.15 | 0.97 |
| | Hanoi 3 | 11.16(2.06) | 9.81(3.01) | -2.07 | 0.65 |
| | Hanoi 4 | 8.16(3.73) | 8.55(3.28) | 0.43 | 0.96 |
| | TMT-A | 82.47(16.56) | 79.70(20.05) | -0.59 | 0.96 |
| | TMT-B | 74.16(13.60) | 69.37(14.79) | -1.32 | 0.87 |
| | Stroop interference | 42.58(19.37) | 48.88(24.42) | 1.15 | 0.90 |

RCF: Rey–Osterrieth complex figure; PA: verbal paired associates; TMT: Trail Making Test.

**Table 2. Visual short-term memory binding test results for both groups.**

| VSTMBT | Risk | Control | t value | *p*-FDR |
|---|---|---|---|---|
| | Mean (SD) | Mean (SD) | | |
| STM Single 2 | 98.25(3.55) | 96.51(6.57) | -1.33 | 0.87 |
| STM Single 3 | 92.69(8.12) | 87.01(19.29) | -1.06 | 0.94 |
| STM Binding 2 | 91.99(9.03) | 89.80(10.99) | -0.87 | 0.96 |
| STM Binding 3 | 71.67(10.07) | 75.67(13.35) | 1.33 | 0.87 |

STM: short-term memory; 2: two objects to remember; 3: three objects to remember.

significant differences in traditional neuropsychological test scores or highly demanding cognitive test scores (supporting information S2–S5 Tables).

   **Regression analysis between the zAP (theta) value and cognition.** The principal component analysis of the *zAP (theta)* revealed three main components: PC1: frontal component (F7, Fp1, Fp2, Fz, F3, F4, F8, and T3); PC2: centroparietal component (Pz, Cz, C4, C3, P4, and P3); and PC3: occipitotemporal component (O1, O2, T6, and T5). No regression model was statistically significant for any of the cognitive scores (Table 4).

## Discussion

In this study, we aimed to explore potential cognitive differences between two groups: individuals with an EEG risk of cognitive impairment, defined by excessive theta activity, and control individuals with a normal EEG. Given the inability to identify cognitive differences between these groups using low-demand tests, our main objective, similar to that of previous studies [16,18,21,22], was to identify potential cognitive alterations in older adults in the presumably preclinical phase of

**Table 3. Loewenstein–Acevedo Scale for semantic interference and learning scores for both groups.**

| LASSI | Risk | Control | t value | p-FDR |
|---|---|---|---|---|
| | Mean (SD) | Mean (SD) | | |
| Free recall A | 10.27(2.66) | 10.85(1.87) | 0.94 | 0.96 |
| Cued recall A | 11.55(2.43) | 12.14(1.48) | 1.11 | 0.90 |
| Cued recall A2 (maximum storage) | 13.97(1.13) | 14.40(0.63) | 1.72 | 0.79 |
| Free recall B | 8.16(2.19) | 8.11(2.02) | -0.13 | 0.98 |
| Cued recall B (PSI) | 8.22(2.49) | 8.59(2.30) | 0.58 | 0.96 |
| Cued recall B2 (r-PSI) | 1.91(1.88) | 2(1.41) | 0.14 | 0.97 |
| Free recall A3 | 8.13(2.90) | 8.59(2.62) | 0.61 | 0.96 |
| Cued recall A3 (RSI) | 9.77(2.86) | 10.11(3.40) | 0.40 | 0.96 |
| Delayed recall A and B | 21.13(4.22) | 21.66(4.11) | 0.46 | 0.96 |
| % of intrusion errors on cued recall B | 0.14(0.11) | 0.14(0.11) | -0.53 | 0.96 |
| % of intrusion errors on cued recall B2 | 0.12(0.12) | 0.14(0.10) | 0.09 | 0.98 |
| Total intrusions | 1.48(1.54) | 1.56(1.76) | 0.44 | 0.96 |

PSI: proactive semantic interference; r-PSI: recovery from proactive semantic interference; RSI: retroactive semantic interference, A: first learning of the first list; A2: second learning of the first list with a cued recall; B: first learning of the second list; B2: second learning of the second list with a cued recall; A3: recall of the first list after interference of list B.

a neurocognitive disorder using highly demanding tests (i.e., VSTMBT and LASSI-II) and traditional neuropsychological tests. Furthermore, our study's strength is that it addresses an important gap in the literature by examining the relationship between EEG-based biomarkers of cognitive impairment risk and sensitive neuropsychological tests.

For the traditional neuropsychological tests, we did not find significant differences between the groups when separating a sample of healthy older adults according to their EEG (excessive theta or normal). The risk group did not differ from the control group, which is consistent with previous results [12–14], despite separating the risk group according to severity (low risk, high risk) or looking for relationships between theta activity and performance in these tests via multiple regression models.

Differences between the risk and control groups were expected when high cognitive demand tests were used; however, we did not detect any significant differences in the present study. Furthermore, we found no significant differences in highly demanding test results between the control group and the low- and high-risk groups. In addition, there was no association between theta activity and the LASSI-II and VSTMBT scores in the regression analysis. Notably, none of the previous studies [16,18,21,22] used EEG as a group formation criterion but instead used clinical and neuropsychological criteria. To our knowledge, the study by Prichep et al. [7] is the only report exploring neuropsychological differences between older adults with and without electroencephalographic risk for cognitive decline that reported cognitive differences, specifically in the Digit Span and Digit Symbol Substitution tests. These contrasting findings might be because the participants in the study mentioned above, in addition to excessive theta AP, had excessive delta AP and reduced beta relative power. The difference may also be due to the health differences between Prichep's participants and our participants; as opposed to Prichep et al., we included only participants who had normal levels of cholesterol, glucose, triglycerides, hemoglobin, and thyroid-stimulating hormone, since those physiological parameters are known to impact brain function [43–46].

Excessive theta AP has been reported to predict cognitive decline 7–10 years later. The subjects in the present study may be at a preclinical phase in which cognitive difficulties can be compensated for, even though brain function is already compromised, especially because the included subjects were 55 years of age and older and because the sociodemographic characteristics of the sample presented a bias toward higher education. Given that educational level is considered a proxy of cognitive reserve [47], the negative results obtained could be due to a protective or compensatory effect on cognitive capacity even when neurophysiological alterations have already occurred. A question that remains pending

future studies is the impact that education level could have on the predictive accuracy of theta AP as a biomarker of the risk of future cognitive decline. Moreover, the impact of other proxies of cognitive reserve, such as verbal intelligence and leisure activities, should be studied, as well as the neurophysiological substrate of the proxies mentioned above, as this could deepen our understanding of how cognitive reserve mitigates the effects of neurophysiological decline and could aid in developing targeted interventions. Furthermore, not only cognitive compensatory strategies but also neurophysiological compensatory mechanisms may be involved in adequately performing cognitive tests. Resting-state EEG does not capture task-specific neural activity associated with cognitive processing or the compensatory mechanisms needed to

**Table 4. Regression analysis results.**

| Cognitive test | $R^2$ | F-Statistic | $p$-FDR |
| --- | --- | --- | --- |
| | | df(3,59) | |
| **RCF** | | | |
| Copy | 0.04 | 1.01 | 0.96 |
| Retrieval | 0.03 | 0.61 | 0.96 |
| **Logical memory** | | | |
| Encoding | 0.02 | 0.39 | 0.96 |
| Retrieval | 0.009 | 0.18 | 0.97 |
| **PA** | | | |
| Encoding | 0.05 | 1.05 | 0.96 |
| Retrieval | 0.009 | 0.17 | 0.97 |
| **Semantic fluency** | 0.02 | 0.43 | 0.96 |
| **Phonologic fluency** | 0.12 | 2.78 | 0.65 |
| **Maze planning** | 0.08 | 1.71 | 0.87 |
| **Hanoi 3** | 0.05 | 1.11 | 0.96 |
| **Hanoi 4** | 0.07 | 1.48 | 0.89 |
| **TMT-A** | 0.02 | 0.50 | 0.96 |
| **TMT-B** | 0.02 | 0.58 | 0.96 |
| **Stroop interference** | 0.03 | 0.76 | 0.96 |
| **STM Single 2** | 0.31 | 0.63 | 0.96 |
| **STM Single 3** | 0.01 | 0.39 | 0.96 |
| **STM Binding 2** | 0.06 | 1.40 | 0.89 |
| **STM Binding 3** | 0.08 | 1.84 | 0.87 |
| **Free recall A** | 0.03 | 0.63 | 0.96 |
| **Cued recall A** | 0.01 | 0.31 | 0.97 |
| **Cued recall A2 (maximum storage)** | 0.07 | 1.55 | 0.89 |
| **Free recall B** | 0.03 | 0.68 | 0.96 |
| **Cued recall B (PSI)** | 0.02 | 0.42 | 0.96 |
| **Cued recall B2(r-PSI)** | 0.01 | 0.38 | 0.96 |
| **Free recall A3** | 0.01 | 0.34 | 0.97 |
| **Cued recall A3(RSI)** | 0.009 | 0.18 | 0.97 |
| **Delayed recall** | 0.09 | 2.08 | 0.87 |
| **% of intrusion errors on cued recall B** | 0.001 | 0.01 | 1.00 |
| **% of intrusion errors on cued recall B2** | 0.003 | 0.06 | 1.00 |
| **Total intrusions** | 0.04 | 0.84 | 0.96 |

RCF: Rey–Osterrieth complex figure; PA: verbal paired associates; TMT: Trail Making Test; STM: short-term memory; r-PSI: recovery from proactive semantic interference; RSI: retroactive semantic interference.

carry out it, as other techniques (EEG, ERP, or fMRI) that explore brain functioning while solving a cognitive task are able to do. For example, neurophysiological compensation could rely on bilateral recruitment of brain regions necessary for task completion, as outlined in the HAROLD model [48], or with different brain network dynamics during task resolution. In fact, Sánchez-Moguel et al. [14] demonstrated that older adults with excessive theta AP had different neurophysiological responses during the Stroop task (no N500 effect) than healthy older adults did, even though the behavioral scores of both groups could not be distinguished. Although differences between these groups have been explored via ERPs during a Stroop task [14] and a syntactic processing task mediated by working memory [13], future studies should evaluate the possibility of changes in ERPs in older adults at the preclinical stage of a potential neurocognitive disorder while high-demand cognitive tests are being performed. This could result in the identification of a very early EEG biomarker for cognitive decline, which would be in line with the ERP alterations reported by Pietto et al. [40] in patients with mild cognitive impairment compared with controls while the VSTMBT was being solved.

These results highlight the advantage of assessing the risk of cognitive decline in older adults via EEG, as it is more sensitive to early changes in brain function than are cognitive–behavioral assessments, regardless of cognitive compensatory mechanisms.

As Rentz et al. [19] established, our findings support the argument that transversal neuropsychological assessments lack sufficient sensitivity to detect subtle cognitive alterations in the preclinical phase of a neurocognitive disorder. It is therefore advised to prioritize the study of neurophysiological markers of risk for cognitive decline with cost-effective instruments such as EEG, including at rest and during task performance (ERPs), which could detect alterations even earlier than neuropsychological tests can.

One of the limitations of this study is the lack of use of established biomarkers in CSF, PET, or genotyping, which could confirm that the sample subjects indeed have a predisposition toward neurodegenerative diseases that result in memory impairments, which could have strengthened our results and could cross-validate EEG findings. Despite having controlled for some confounding variables that may influence the EEG (cholesterol, glucose, triglyceride, hemoglobin, and thyroid-stimulating hormone), future studies should also consider sleep hours [49], diet [50], and exercise regime [51] because they have been shown to alter the resting-state EEG. Further limitations include a small sample size and a bias toward higher education, which may limit the statistical power for detecting subtle differences and the generalizability of our results, respectively. Another limitation is that we did not assess cognitive deficits using all available cognitive tests for neuropsychological testing or computerized cognitive assessment, which could provide more precise measurements (e.g., reaction times) and, thus, may detect subtle impairments. Future studies should expand on our results by using other neuropsychological tests and methodologies (i.e., longitudinal testing) to corroborate our findings. Furthermore, a follow-up retest in a 2-year time-frame of our cohort sample would be convenient to further assess the predictive accuracy of excessive theta AP on cognitive decline and its correlation with performance on high-demand and traditional neuropsychological tests.

Even though our research is primarily based on excess theta AP as a risk biomarker for cognitive decline, this is not the only reported EEG risk predictor of cognitive impairment. Prichep et al. [7] established a predictive model of future cognitive decline that included a mix of the mean frequency of the total spectrum on C4, the mean frequency in the delta on T3T5 and P3O1, the AP across all frequency bands diffusely on the right hemisphere and more so on F8, and the theta AP on FP2, F4, and F8. Furthermore, higher delta power, lower alpha power, and lower alpha peak frequency have been associated with clinical progression from preclinical to dementia status [11]. In this study, we compared groups for AP in other frequency bands and found no significant differences. We chose the theta AP excess as a predictor because it is the most deviant from the norms [7] and because of its straightforward implementation in the clinical setting.

Moreover, cognitive reserve, cognitive performance, and brain biomarkers appear to exhibit a nonlinear relationship [52]. Thus, using multidimensional models would be an interesting avenue for future research; for example, machine learning approaches could identify latent patterns across theta power, cognitive test scores, and demographic variables, which could improve predictive precision.

In conclusion, the urgency of early detection of pathologic cognitive decline cannot be overlooked. Early diagnosis of a neurodegenerative disease and subsequent early treatment can significantly improve the quality of life of patients. Our results suggest that neurophysiological techniques can detect subtle changes in brain function sooner than neuropsychological assessments can, even when high-demand cognitive tests such as the VSTMBT and the LASSI-II are administered, underscoring the significance of our research. Therefore, it may be desirable to include electroencephalographic recordings in routine neurological assessments of older adults to identify those who are at risk for cognitive impairment. Future investigations should study the relationship between EEG alterations and cognitive decline in longitudinal follow-ups, reinforcing the importance of early interventions and treatments.

## Supporting information

**S1 Table. Risk *vs*. Control group differences on the WAIS-IV.**
(DOCX)

**S2 Table. High risk *vs.* Low risk *vs.* Control group differences on the median of the sum of zAP(theta).**
(DOCX)

**S3 Table. High risk *vs.* Low risk *vs.* Control group differences on the zAP(theta) average.**
(DOCX)

**S4 Table. High risk *vs.* Low risk *vs.* Control group differences on the sum of electrodes with excess zAP(theta).**
(DOCX)

**S5 Table. High risk *vs.* Low risk *vs.* Control group differences on the sum of zAP(theta) squared (maintaining the sign).**
(DOCX)

## Acknowledgments

The authors would like to express their gratitude to Thalia Harmony Baillet, Héctor Belmont, Paloma Arlet Roa Rojas, Luisa Mariana Pérez Figueroa, Teresa Álvarez, Gina Lorena Quirarte, Nuri Aranda López, Daniela Roldán García, Bertha Esquivel, and Carolina Villada for their technical support. This article constitutes a part of Jorge Sigg-Alonso's PhD project.

## Author contributions

**Conceptualization:** Jorge Sigg-Alonso, Mauricio González-López, Thalía Fernández.

**Data curation:** Jorge Sigg-Alonso.

**Formal analysis:** Jorge Sigg-Alonso.

**Funding acquisition:** Thalía Fernández.

**Investigation:** Jorge Sigg-Alonso, Mauricio González-López, Eduardo Gonzalez-Moreira, Rodolfo Solís-Vivanco, Thalía Fernández.

**Methodology:** Jorge Sigg-Alonso, Mauricio González-López, Rodolfo Solís-Vivanco, Thalía Fernández.

**Project administration:** Thalía Fernández.

**Resources:** Thalía Fernández.

**Software:** Jorge Sigg-Alonso, Eduardo Gonzalez-Moreira.

**Supervision:** Thalía Fernández.

**Visualization:** Jorge Sigg-Alonso.

**Writing – original draft:** Jorge Sigg-Alonso.

**Writing – review & editing:** Jorge Sigg-Alonso, Mauricio González-López, Eduardo Gonzalez-Moreira, Rodolfo Solís-Vivanco, Thalía Fernández.

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
