## [Decision Letter · Decision Letter 0]

12 Nov 2024

Dear Dr. Fernández,

Thank you for submitting your manuscript to PLOS ONE. After careful consideration, we feel that it has merit but does not fully meet PLOS ONE’s publication criteria as it currently stands. Therefore, we invite you to submit a revised version of the manuscript that addresses the points raised during the review process.

We look forward to receiving your revised manuscript.

Kind regards,

Gabriel G. De La Torre

Academic Editor

PLOS ONE

**Journal Requirements:**

This work was supported by grants IN225414, IN200817, and IT201123 from the Universidad Nacional Autónoma de México (DGAPA-PAPIIT), with TF as recipient. 

During this study, Jorge A. Sigg-Alonso was a beneficiary of scholarship number 765024 by Consejo Nacional de Humanidades, Ciencia y Tecnología (CONAHCYT).

3. Please note that your Data Availability Statement is currently missing the repository name. If your manuscript is accepted for publication, you will be asked to provide these details on a very short timeline. We therefore suggest that you provide this information now, though we will not hold up the peer review process if you are unable.

Reviewers' comments:

Reviewer's Responses to Questions

**Comments to the Author**

1. Is the manuscript technically sound, and do the data support the conclusions?

Reviewer #1: Partly

Reviewer #2: Yes

2. Has the statistical analysis been performed appropriately and rigorously?

Reviewer #1: Yes

Reviewer #2: Yes

3. Have the authors made all data underlying the findings in their manuscript fully available?

Reviewer #1: Yes

Reviewer #2: Yes

4. Is the manuscript presented in an intelligible fashion and written in standard English?

Reviewer #1: Yes

Reviewer #2: Yes

**Reviewer #1: ** This study investigated whether healthy older adults with excessive theta absolute power (AP) in their EEG, considered at risk for cognitive impairment, show reduced performance on highly demanding cognitive tasks. The researchers compared performance on the Visual Short-Term Memory Binding Test (VSTMBT) and the Loewenstein-Acevedo Scale for Semantic Interference and Learning (LASSI-II) between older adults with and without excessive theta AP. The results showed that no significant differences were found between the risk group (excessive theta AP) and control group on any test scores from the VSTMBT or LASSI-II. The degree of excessive theta AP did not predict performance on these highly demanding cognitive tasks.

There were no significant differences in performance when comparing high-risk, low-risk, and control groups. Traditional neuropsychological tests also showed no significant differences between groups.

The authors conclude that older adults with excessive theta AP do not exhibit impaired performance in highly demanding cognitive contexts, despite being at electroencephalographic risk of developing cognitive impairment in the future.

The study addresses an important gap in the literature by examining the relationship between EEG-based biomarkers of cognitive impairment risk and sensitive neuropsychological tests.

The use of both traditional and novel, highly demanding cognitive tests provides a comprehensive assessment. The inclusion of multiple criteria for defining risk groups adds robustness to the analysis. However, to my point of view, there are still some limitations need to be further addressed:

The statistical approaches, including permutation testing,PCA and multiple comparison corrections, are appropriate and robust. However, the sample size is relatively small, which may limit statistical power to detect subtle differences.

The cross-sectional design prevents conclusions about the predictive value of excessive theta AP for future cognitive decline.

The lack of significant differences between groups is intriguing and warrants further investigation. The authors' discussion of cognitive reserve as a potential explanation is appropriate.

It would be valuable to consider other factors that might influence the relationship between EEG markers and cognitive performance, such as lifestyle factors or genetic risk.

Overall, this is a well-conducted study that makes a valuable contribution to the field of cognitive aging and EEG biomarkers. While the negative findings are yielded, the authors are suggested to provide more detailed results interpretation with important insights and directions for future research.

**Reviewer #2: ** Peer reviewed by Jacob A. Eder., PhD,

For manuscript submission and further recommendations for conciseness of the scope of study.

PLOS ONE

No evidence of reduced capacity during highly demanding cognitive tasks in healthy

older adults at electroencephalographic risk of cognitive impairment

--Manuscript Draft—

PONE-D-24-42123

Article Type: Research Article

Full Title: No evidence of reduced capacity during highly demanding cognitive tasks in healthy

older adults at electroencephalographic risk of cognitive impairment

Short Title: Electroencephalographic risk of cognitive impairment and cognition

Corresponding Author: Thalía Fernández, Ph.D.

Universidad Nacional Autonoma de Mexico

Querétaro, Querétaro MEXICO

The manuscript demonstrates methodological rigor suitable for PLOS ONE publication standards, based on its design, analysis, and scope of inquiry into EEG biomarkers and cognitive assessment outcomes. The study implements a longitudinal and correlational approach, examining EEG-based markers in tandem with high-demand cognitive assessments to assess risk factors associated with cognitive decline in older adults—a population that often falls within the scope of neurodegenerative studies.

The participant inclusion criteria are meticulously outlined, encompassing individuals within the specified age range who meet neuropsychological benchmarks indicative of healthy cognitive function. Through the application of electroencephalography (EEG) with a specific focus on theta power as a predictive biomarker, the methodology aligns with established neurophysiological protocols. Notably, the use of non-invasive EEG as a predictive tool allows for a nuanced exploration of cognitive reserve and decline, serving as a non-intrusive alternative to cerebrospinal biomarkers, which carry significant procedural risks and costs.

From a neuropsychological perspective, the use of high-cognitive-demand tests—namely, the Visual Short-Term Memory Binding Test (VSTMBT) and the Loewenstein-Acevedo Scale for Semantic Interference and Learning (LASSI-II)—demonstrates an attempt to unveil cognitive decline undetectable by conventional neuropsychological assessments. Both assessments are neuropsychometrically validated tools sensitive to early cognitive dysfunction, and their inclusion reflects an understanding of the need for advanced diagnostic tools in clinical neuropsychology.

Statistical analysis is executed with precision through permutation-based testing, principal component analysis (PCA), and regression modelling, providing a robust framework to address the potential predictive value of theta power. Corrective measures for multiple comparisons, including false discovery rate (FDR) adjustments, reflect a statistical approach that minimizes Type I error risk, thereby strengthening the validity of the findings.

Although this study’s results do not show significant cognitive differences between the groups in either the high-demand or conventional assessments, the findings contribute to the body of literature by challenging the sensitivity of EEG-based markers to predict cognitive deterioration in cognitively healthy older adults. This conclusion aligns with the notion of cognitive reserve, suggesting that neurodegeneration may not immediately manifest behaviourally in individuals with compensatory cognitive mechanisms. Further investigation could explore the potential protective role of higher education levels, which might contribute to these individuals' apparent cognitive resilience.

In summary, the study exemplifies adherence to PLOS ONE’s publication criteria, utilizing quantifiable, reproducible, and rigorous techniques. The limitations acknowledged by the authors, such as the educational bias and restricted generalizability, are pertinent and recommend avenues for subsequent research to employ broader, demographically diverse sampling to validate these findings. This study thus establishes a foundation for expanding the application of EEG biomarkers in predictive neuropsychological assessments of age-related cognitive decline.

The study, while methodologically sound, would benefit from addressing several key areas to strengthen its contributions to neuropsychological and neurophysiological literature on age-related cognitive decline. Expanding upon these areas in future research would enhance both the study’s impact and its applicability within broader clinical and research contexts. The following recommendations are advised:

1. Broader Biomarker Integration:

The study currently relies solely on theta power as the electroencephalographic marker, which, while valuable, may not capture the full range of neurodegenerative processes linked to cognitive impairment. Integrating additional EEG frequency bands (e.g., alpha, delta, beta) could provide a more comprehensive profile of neurophysiological function, particularly in preclinical populations. Furthermore, incorporating other biomarkers, such as cerebrospinal fluid (CSF) amyloid-beta and tau levels, or neuroimaging markers like PET scans for amyloid or tau deposition, would enable a multi-modal approach that could cross-validate EEG findings and strengthen predictive accuracy.

2. Longitudinal Monitoring:

Although, this study provides a snapshot of cognitive and EEG measures, longitudinal monitoring over extended periods would reveal whether excessive theta power correlates with cognitive decline over time in the same individuals. Such an approach would enhance understanding of the temporal dynamics of theta power and its potential role as an early marker for neurocognitive disorders. Tracking changes in theta activity and cognitive performance over intervals (e.g., every six months to one year) would help delineate the predictive timeline for cognitive impairment in aging populations.

3. Diverse and Representative Sample:

This study’s sample shows an educational bias toward highly educated participants, potentially contributing to a heightened cognitive reserve that could mitigate detectable neuropsychological impairment. Future studies should aim for a sample that includes a diverse range of educational backgrounds, as well as socio-demographic diversity, to improve the generalizability of findings. It would be particularly informative to examine whether theta power’s predictive value varies across individuals with different baseline cognitive reserves.

4. Use of Additional Cognitive Assessments:

While the Visual Short-Term Memory Binding Test (VSTMBT) and Loewenstein-Acevedo Scale for Semantic Interference and Learning (LASSI-II) are suitable for assessing high-cognitive demand and are sensitive to subtle deficits, a broader neuropsychological battery would capture a wider array of cognitive functions potentially affected by neurodegenerative processes. Including tests that assess executive functions, attention, and processing speed—such as the Stroop Task, Trail Making Test, and Wisconsin Card Sorting Test—could reveal more nuanced cognitive impairments associated with excessive theta power. Additionally, the use of computerized cognitive assessments could facilitate finer-grained analysis of reaction times and error patterns, yielding a more detailed cognitive profile.

5. Exploration of Compensatory Mechanisms and Cognitive Reserve:

The study suggests that high educational attainment may serve as a proxy for cognitive reserve, potentially enabling individuals to maintain cognitive performance despite neurophysiological alterations. Further exploration of compensatory mechanisms through functional neuroimaging (e.g., fMRI or MEG during task performance) could clarify the neural correlates of cognitive reserve in older adults with excessive theta power. Identifying brain regions or networks that exhibit compensatory activation patterns would deepen the understanding of how cognitive reserve mitigates the effects of neurophysiological decline and could aid in developing targeted interventions.

6. Enhanced EEG Protocols and Dynamic Testing Conditions:

Given that EEG was recorded in a resting state, it may not fully capture task-specific neural activity variations that emerge during cognitive engagement. Employing task-based EEG recordings, particularly during high-cognitive-demand tasks like the VSTMBT and LASSI-II, would yield data on event-related potentials (ERPs) and dynamic EEG responses associated with cognitive performance. ERPs, particularly within theta, alpha, and gamma frequencies, could reveal task-specific deficits not apparent in resting-state EEG and provide a more detailed understanding of cognitive processing under demanding conditions.

7. Data Sharing and Open Access:

In line with open science practices, making EEG datasets and cognitive performance scores publicly available (where ethical and permissible) would enable independent verification and meta-analytic integration of findings across studies. While the data availability statement mentions restrictions on the LASSI-II data, public accessibility of non-restricted datasets (e.g., EEG data and VSTMBT scores) would facilitate further research and contribute to building a cumulative evidence base for EEG markers in cognitive impairment.

8. Consideration of Potential Confounding Factors:

Factors such as cardiovascular health, diet, and sleep quality—known to impact both EEG measures and cognitive function—are not accounted for in the current analysis. Future studies should consider controlling for or assessing these variables, as they could interact with theta power and cognitive performance. Screening for cardiovascular markers, diet quality, or incorporating self-reported sleep assessments could account for potential confounding effects and provide a more refined understanding of theta power’s relationship with cognitive health.

9. Investigating Non-Linear and Interaction Effects:

The current analysis appears to focus on linear relationships between theta power and cognitive performance. Wherever, cognitive decline may not exhibit a strictly linear association with theta power, particularly in populations with high cognitive reserve. Future research could investigate non-linear models or explore interactions between theta power and other demographic or neuropsychological variables (e.g., age, education level, and baseline cognitive performance) to uncover complex patterns that linear models may miss.

10. Developing a Predictive Model with Machine Learning:

Machine learning (ML) techniques, such as random forests or support vector machines, could be employed to develop predictive models using the study’s EEG and neuropsychological data. ML approaches can handle multi-dimensional data and potentially identify latent patterns across theta power, cognitive test scores, and demographic variables, which could improve predictive precision. ML algorithms may thus offer a practical solution for clinicians to assess cognitive risk factors using EEG data, integrating it into a tool for early-stage dementia risk assessment.

Collectively, these recommendations would elevate the study's contributions to the field, offering a more comprehensive understanding of the interplay between EEG biomarkers, cognitive reserve, and neuropsychological assessment in older adults. By addressing these suggestions, future research would refine EEG-based cognitive risk assessment tools, aligning with broader clinical and scientific initiatives for the early detection of neurodegenerative disorders.

RECOMMENDATIONS TO CONSIDER BEFORE PUBLICATION FINALISATION

Based on the thorough examination of this study, the manuscript demonstrates a high level of methodological rigor and scientific integrity, making it a strong candidate for publication. Wherever, while the study aligns well with the standards of PLOS ONE, appropriately and professionally incorporating specific revisions before submission could enhance the manuscript’s robustness and potential impact. Here is a summary of suggestions to consider:

1. Expanded Biomarker Assessment: While theta power is valuable, adding a discussion on the limitations of relying on a single EEG band as a marker and suggesting future studies include additional biomarkers would present a more comprehensive analysis. Mentioning plans to integrate multi-modal biomarker approaches, even if not implemented here, would position this work within the broader context of cognitive impairment research.

2. Longitudinal Perspective Emphasis: Reinforcing the potential of a longitudinal follow-up in a future study, particularly considering theta power's predictive window, would give readers a sense of the study’s next logical steps. While this is currently a cross-sectional study, emphasizing the importance of observing cognitive and theta power changes over time would underscore its contribution to the field and encourage replication studies.

3. Task-Based EEG Recommendation: The manuscript would benefit from acknowledging the limitation of using resting-state EEG alone and suggesting task-based EEG or event-related potential (ERP) testing in future studies. Briefly discussing how task-based EEG might reveal distinct neural activity patterns associated with cognitive effort could add depth to the methodological context.

4. Broader Sample Demographics: Addressing the educational homogeneity of the sample more directly, with an acknowledgment of its potential impact on cognitive reserve, could enhance transparency. Proposing a more demographically diverse sample in future studies, as a brief note in the limitations, would also demonstrate attentiveness to generalizability issues.

5. Confounding Variables: A short discussion on the influence of confounding factors such as diet, cardiovascular health, or sleep quality, while not feasible within the current scope, would provide context for the findings and inform future studies seeking to disentangle EEG changes specific to cognitive impairment from those influenced by lifestyle factors.

If these suggestions are integrated concisely, the manuscript will present a well-rounded perspective that addresses the current findings while inviting future exploration. Overall, the paper is near ready for submission, but these minor adjustments would strengthen its reception and significance within the field.

**Do you want your identity to be public for this peer review?** For information about this choice, including consent withdrawal, please see our Privacy Policy

Reviewer #1: **Yes: ** Bochao Lin

Reviewer #2: **Yes: ** Jacob A. Eder, PhD

---

## [Author Response · Author response to Decision Letter 0]

11 Dec 2024

Dear Dr. Gabriel G. De La Torre,

We are thankful for this first set of revisions provided by the referees. We feel our paper has been remarkably improved by the referees’ suggestions. We provide our responses to each reviewer separately:

General Notes:

I. Tables in the manuscript and the supporting information were reformatted to adhere to PLOS ONE specifications. Authors’ contributions were delineated.

II. We included the statement “There was no additional external funding received for this study” in the updated Funding Statement.

III. We will make our data publicly available in the Github repository: https://github.com/JorgeSigg/No-evidence-of-reduced-capacity-during-highly-demanding-cognitive-tasks-in-older-adults-at-EEG-risk-

IV. We double checked and corrected the reference list.

V. The reviewers' suggestions were taken into account.

Responses to Reviewer 1:

R1.1.- The study addresses an important gap in the literature by examining the relationship between EEG-based biomarkers of cognitive impairment risk and sensitive neuropsychological tests.

A: Thank you for your comment, we agree that there is a need for addressing this gap. We included your comment as a strength of our study (line 305 - 307): Furthermore, our study's strength is that it addresses an important gap in the literature by examining the relationship between EEG-based biomarkers of cognitive impairment risk and sensitive neuropsychological tests.

R1.2.- The use of both traditional and novel, highly demanding cognitive tests provides a comprehensive assessment. The inclusion of multiple criteria for defining risk groups adds robustness to the analysis. However, to my point of view, there are still some limitations need to be further addressed:

R1.2.1- The statistical approaches, including permutation testing, PCA and multiple comparison corrections, are appropriate and robust. However, the sample size is relatively small, which may limit statistical power to detect subtle differences.

A: Thank you for your comment. We took your advice and stated explicitly this as a limitation of the study (lines 372 - 374): Further limitations include a small sample size and a bias toward higher education, which may limit the statistical power for detecting subtle differences and the generalizability of our results, respectively.

R1.2.2- The cross-sectional design prevents conclusions about the predictive value of excessive theta AP for future cognitive decline.

A: We are sorry for not being sufficiently clear about this. Predictive studies with a longitudinal design were done by Prichep et al. (2006) and Gouw et al. (2017), with a followed up of 10 years and 2 years, respectively. In the present study we considered the established predictive value of theta excess, which was demonstrated by the aforementioned authors. Assuming the predictive value of the excess theta, our study compares two groups of healthy older adults with cognition within normal limits in traditional tests: one group with excess theta (risk group) vs. a group with normal EEG (control), to explore if there are cognitive differences in more demanding cognitive tests between both groups. Adjustments were made in the introduction for clarification purposes (94 - 98): Our study breaks new ground in this field by investigating the potential relationship between EEG-based biomarkers of cognitive impairment risk and neuropsychological tests that have demonstrated high sensitivity to the preclinical phases of neurocognitive disorders. Considering the predictive value of excessive theta activity for cognitive decline, the aim of this study was to fill a significant gap in the current literature.

We are aware of the value of a follow-up assessment of this cohort and added a comment (lines 379 - 382): Furthermore, a follow-up retest in a 2-year timeframe of our cohort sample would be convenient to further assess the predictive accuracy of excessive theta AP on cognitive decline and its correlation with performance on high-demand and traditional neuropsychological tests.

R1.2.3.- The lack of significant differences between groups is intriguing and warrants further investigation. The authors' discussion of cognitive reserve as a potential explanation is appropriate.

A: We are grateful for your comment, and we agree that the lack of cognitive differences in more demanding tasks is most intriguing.

R1.2.4.- It would be valuable to consider other factors that might influence the relationship between EEG markers and cognitive performance, such as lifestyle factors or genetic risk.

A: We are in agreement that there are other factors that influence this relationship, and in other articles we have published the impact that physical activity (Sanchez-Lopez et al., 2018) and cognitive reserve (Ferrari-Díaz et al., 2022) have on this relationship. In fact, we are currently gathering blood samples to establish if there is a relationship between this EEG predictor (excess theta) and genetic (apoE-ε4) and inflammatory (IL-6) risk factors. In the manuscript we made more explicit this study’s limitations (366 - 372): One of the limitations of this study is the lack of use of established biomarkers in CSF, PET, or genotyping, which could confirm that the sample subjects indeed have a predisposition toward neurodegenerative diseases that result in memory impairments, which could have strengthened our results and could cross-validate EEG findings. Despite having controlled for some confounding variables that may influence the EEG (cholesterol, glucose, triglyceride, hemoglobin, and thyroid-stimulating hormone), future studies should also consider sleep hours [49], diet [50], and exercise regime [51] because they have been shown to alter the resting-state EEG.

Responses to Reviewer 2:

R2.1.- Broader Biomarker Integration: The study currently relies solely on theta power as the electroencephalographic marker, which, while valuable, may not capture the full range of neurodegenerative processes linked to cognitive impairment. Integrating additional EEG frequency bands (e.g., alpha, delta, beta) could provide a more comprehensive profile of neurophysiological function, particularly in preclinical populations.

A: Thank you for your comments. We incorporated a paragraph in the discussion section about other frequency bands and their relationship to cognitive decline (383 - 392): Even though our research is primarily based on excess theta AP as a risk biomarker for cognitive decline, this is not the only reported EEG risk predictor of cognitive impairment. Prichep et al. [7] established a predictive model of future cognitive decline that included a mix of the mean frequency of the total spectrum on C4, the mean frequency in the delta on T3T5 and P3O1, the AP across all frequency bands diffusely on the right hemisphere and more so on F8, and the theta AP on FP2, F4, and F8. Furthermore, higher delta power, lower alpha power, and lower alpha peak frequency have been associated with clinical progression from preclinical to dementia status [52]. In this study, we compared groups for AP in other frequency bands and found no significant differences. We chose the theta AP excess as a predictor because it is the most deviant from the norms [7] and because of its straightforward implementation in the clinical setting.

R2.2.- Longitudinal Monitoring: Although, this study provides a snapshot of cognitive and EEG measures, longitudinal monitoring over extended periods would reveal whether excessive theta power correlates with cognitive decline over time in the same individuals. Such an approach would enhance understanding of the temporal dynamics of theta power and its potential role as an early marker for neurocognitive disorders. Tracking changes in theta activity and cognitive performance over intervals (e.g., every six months to one year) would help delineate the predictive timeline for cognitive impairment in aging populations.

A: Longitudinal studies are indeed very valuable. We are sorry for not being sufficiently clear about this. Predictive studies with a longitudinal design were done by Prichep et al. (2006) and Gouw et al. (2017), with a followed up of 10 years and 2 years, respectively. In the present study we considered the established predictive value of theta excess, which was demonstrated by the aforementioned authors. Assuming the predictive value of the excess theta, our study compares two groups of healthy older adults with cognition within normal limits in traditional tests: one group with excess theta (risk group) vs. a group with normal EEG (control), to explore if there are cognitive differences in more demanding cognitive tests between both groups. Adjustments were made in the introduction for clarification purposes (94 - 98): Our study breaks new ground in this field by investigating the potential relationship between EEG-based biomarkers of cognitive impairment risk and neuropsychological tests that have demonstrated high sensitivity to the preclinical phases of neurocognitive disorders. Considering the predictive value of excessive theta activity for cognitive decline, the aim of this study was to fill a significant gap in the current literature.

We are aware of the value of a follow-up assessment of this cohort and added a comment (lines 379 - 382): Furthermore, a follow-up retest in a 2-year timeframe of our cohort sample would be convenient to further assess the predictive accuracy of excessive theta AP on cognitive decline and its correlation with performance on high-demand and traditional neuropsychological tests.

R2.3.- Diverse and Representative Sample: This study’s sample shows an educational bias toward highly educated participants, potentially contributing to a heightened cognitive reserve that could mitigate detectable neuropsychological impairment. Future studies should aim for a sample that includes a diverse range of educational backgrounds, as well as socio-demographic diversity, to improve the generalizability of findings. It would be particularly informative to examine whether theta power’s predictive value varies across individuals with different baseline cognitive reserves.

A: We further commented on this topic (lines 338 - 343): A question that remains pending future studies is the impact that education level could have on the predictive accuracy of theta AP as a biomarker of the risk of future cognitive decline. Moreover, the impact of other proxies of cognitive reserve, such as verbal intelligence and leisure activities, should be studied, as well as the neurophysiological substrate of the proxies mentioned above, as this could deepen our understanding of how cognitive reserve mitigates the effects of neurophysiological decline and could aid in developing targeted interventions.

R2.4.- Use of Additional Cognitive Assessments: While the Visual Short-Term Memory Binding Test (VSTMBT) and Loewenstein-Acevedo Scale for Semantic Interference and Learning (LASSI-II) are suitable for assessing high-cognitive demand and are sensitive to subtle deficits, a broader neuropsychological battery would capture a wider array of cognitive functions potentially affected by neurodegenerative processes. Including tests that assess executive functions, attention, and processing speed—such as the Stroop Task, Trail Making Test, and Wisconsin Card Sorting Test—could reveal more nuanced cognitive impairments associated with excessive theta power. Additionally, the use of computerized cognitive assessments could facilitate finer-grained analysis of reaction times and error patterns, yielding a more detailed cognitive profile.

A: We agree that it is a limitation not to include the Wisconsin Card Sorting Test in our neuropsychological battery and not to use computerized tests or virtual reality tests; future studies should incorporate the latter assessment tool for it could detect more subtle differences between groups. We added a comment on this (lines 374 - 377): Another limitation is that we did not assess cognitive deficits using all available cognitive tests for neuropsychological testing or computerized cognitive assessment, which could provide more precise measurements (e.g., reaction times) and, thus, may detect subtle impairments.

R2.5.- Exploration of Compensatory Mechanisms and Cognitive Reserve:

The study suggests that high educational attainment may serve as a proxy for cognitive reserve, potentially enabling individuals to maintain cognitive performance despite neurophysiological alterations. Further exploration of compensatory mechanisms through functional neuroimaging (e.g., fMRI or MEG during task performance) could clarify the neural correlates of cognitive reserve in older adults with excessive theta power. Identifying brain regions or networks that exhibit compensatory activation patterns would deepen the understanding of how cognitive reserve mitigates the effects of neurophysiological decline and could aid in developing targeted interventions.

A: We further discussed this idea (lines 340 - 350): Furthermore, not only cognitive compensatory strategies but also neurophysiological compensatory mechanisms may be involved in adequately performing cognitive tests. Resting-state EEG does not capture task-specific neural activity associated with cognitive processing or the compensatory mechanisms needed to carry out it, as other techniques (EEG, ERP, or fMRI) that explore brain functioning while solving a cognitive task are able to do. For example, neurophysiological compensation could rely on bilateral recruitment of brain regions necessary for task completion, as outlined in the HAROLD model [48], or with different brain network dynamics during task resolution. In fact, Sánchez-Moguel et al. [14] demonstrated that older adults with excessive theta AP had different neurophysiological responses during the Stroop task (no N500 effect) than healthy older adults did, even though the behavioral scores of both groups could not be distinguished.

R2.6.- Enhanced EEG Protocols and Dynamic Testing Conditions:

Given that EEG was recorded in a resting state, it may not fully capture task-specific neural activity variations that emerge during cognitive engagement. Employing task-based EEG recordings, particularly during high-cognitive-demand tasks like the VSTMBT and LASSI-II, would yield data on event-related potentials (ERPs) and dynamic EEG responses associated with cognitive performance. ERPs, particularly within theta, alpha, and gamma frequencies, could reveal task-specific deficits not apparent in resting-state EEG and provide a more detailed understanding of cognitive processing under demanding conditions.

A: We agree that ERP or desynchronization studies while solving high demanding cognitive tests would be interesting research. In the introduction we mentioned 2 ERP studies (Stroop and a syntactic processing task mediated by working memory), in which groups differed in its electrophysiological response. Still, it is important to acknowledge the value that resting EEG has as it reflects the functional integrity of the central nervous system. We added a comment on this (lines 350 - 356): Although differences between these groups have been explored via ERPs during a Stroop task [14] and a syntactic processing task mediated by working memory [13], future studies should evaluate the possibility of changes in ERPs in older adults at the preclinical stage of a potential neurocognitive disorder while high-demand cognitive tests are being performed. This could result in the identification of a very early EEG biomarker for cognitive decline, which would be in line with the ERP alterations reported by Pietto et al. [40] in patients with mild cognitive impairment compared with controls while the VSTMBT was being solved.

R2.7.- Data Sharing and Open Access: In line with open science practices, making EEG datasets and cognitive performance scores publicly available (where ethical and permissible) would enable independent verification and meta-analytic integration of findings across studies. While the data availability statement mentions restrictions on the LASSI-II data, public accessibility of non-restricted datasets (e.g., EEG data and VSTMBT scores) would facilitate further research and

---

## [Decision Letter · Decision Letter 1]

25 Feb 2025

No evidence of reduced capacity during highly demanding cognitive tasks in healthy older adults at electroencephalographic risk of cognitive impairment

PONE-D-24-42123R1

Dear Dr. Fernández,

We’re pleased to inform you that your manuscript has been judged scientifically suitable for publication and will be formally accepted for publication once it meets all outstanding technical requirements.

Kind regards,

Giulia Prete

Academic Editor

PLOS ONE

Additional Editor Comments (optional):

Reviewers are very satisfied with the revisions made. I carefully revised the manuscript myself and I believe it adds important evidence in this research field. Thus, I am happy to support the publication of the manuscirpt in ots present form and I would like to thank very much the expert Reviewers for the time spent in their revisions, and the Authors who ully considered the suggestions received. As you can see below, Reviewer 2 only recommends to further specify future research directions: I leave it to the authors to decide whether and how to implement this point. 

Reviewer's Responses to Questions

**Comments to the Author**

Reviewer #1: (No Response)

Reviewer #2: All comments have been addressed

2. Is the manuscript technically sound, and do the data support the conclusions?

Reviewer #1: Yes

Reviewer #2: Yes

3. Has the statistical analysis been performed appropriately and rigorously?

Reviewer #1: Yes

Reviewer #2: Yes

4. Have the authors made all data underlying the findings in their manuscript fully available?

Reviewer #1: Yes

Reviewer #2: Yes

5. Is the manuscript presented in an intelligible fashion and written in standard English?

Reviewer #1: Yes

Reviewer #2: Yes

Reviewer #1: I have reviewed the manuscript "No evidence of reduced capacity during highly demanding cognitive tasks in healthy

older adults at electroencephalographic risk of cognitive impairment" for the previous submission and recommended revisions. The authors have responded to my queries and incorporated it into the manuscript, along with feedback from other reviewers, which I believe improved the quality of the MS acceptable to be published.

Reviewer #2: Reviewer’s Reassessment of Manuscript PONE-D-24-42123R1

The revised manuscript titled “No evidence of reduced capacity during highly demanding cognitive tasks in healthy older adults at electroencephalographic risk of cognitive impairment” has been carefully evaluated following the incorporation of the recommended revisions. The authors have demonstrated a rigorous and methodical approach in addressing the concerns raised during the initial review, resulting in a significantly refined and robust manuscript.

Strengths of the Revised Manuscript

Comprehensiveness of Methodological Enhancements

The study's methodological framework has been strengthened through clearer articulation of the rationale for group selection, particularly in distinguishing between the risk and control groups based on theta absolute power (AP) z-scores. The authors have also elaborated on the statistical approaches employed, ensuring transparency in the permutation testing, principal component analysis (PCA), and multiple comparison corrections, which add robustness to the analytical rigor of the study.

Clarification of the Study’s Scope and Predictive Framework

The clarification regarding the cross-sectional nature of the study and the explicit reference to prior longitudinal studies (Prichep et al., 2006; Gouw et al., 2017) effectively contextualizes the findings. By acknowledging the limitations of a single-time-point design while reinforcing the predictive validity of theta AP excess through prior evidence, the authors have provided a balanced discussion that situates the present findings within the broader literature.

Refinement of Statistical Limitations and Sample Considerations

The acknowledgment of the study’s relatively small sample size and its implications for detecting subtle cognitive differences has been appropriately incorporated. Additionally, the discussion regarding potential biases introduced by the high level of education within the sample has been expanded, reinforcing the need for future studies to examine diverse educational backgrounds.

Enhanced Discussion on Cognitive Reserve and Compensatory Mechanisms

The manuscript has been notably strengthened by the expanded discussion on cognitive reserve and compensatory neural mechanisms. The references to the HAROLD model and potential bilateral recruitment in older adults exhibiting high theta AP provide a compelling theoretical framework that adds depth to the interpretation of null findings. Furthermore, the inclusion of prior research linking physical activity, inflammation markers, and genetic risk factors to EEG-based cognitive predictions adds a valuable multi-dimensional perspective.

Integration of Broader Neurophysiological and Biomarker Considerations

The discussion of additional EEG frequency bands (e.g., alpha, delta, beta) and their potential role in cognitive decline strengthens the manuscript's neurophysiological breadth. While theta power remains the primary focus, the authors have effectively contextualized its relevance within a broader spectrum of electrophysiological biomarkers.

Commitment to Open Science and Data Transparency

The inclusion of a publicly accessible dataset via the GitHub repository is commendable and aligns with best practices in scientific transparency. The explanation regarding the delayed release of LASSI-II scores until the completion of the normative database provides a reasonable and ethical justification.

Areas for Further Consideration

Potential for Nonlinear Analytical Approaches

While the manuscript primarily explores linear relationships between theta power and cognitive function, cognitive decline is increasingly understood to exhibit nonlinear patterns, particularly in individuals with high cognitive reserve. The suggestion to explore non-parametric or machine learning approaches (e.g., random forests, support vector machines) is well-founded and could be incorporated into future research directions.

Utility of Task-Based EEG Measures

The resting-state EEG approach remains a widely used and valid neurophysiological tool; however, the potential benefits of event-related potentials (ERPs) in elucidating task-specific neural dynamics have been acknowledged. Future studies incorporating cognitive task-based EEG paradigms could yield more granular insights into compensatory mechanisms.

Generalizability and Population-Level Implications

Although the study provides valuable insights into EEG-based risk assessment, the high level of education within the sample poses limitations for generalizability. Additional discussion regarding how these findings translate to populations with lower cognitive reserve or different cultural contexts could enhance the study’s applicability.

Final Assessment

The revisions implemented in response to the initial peer review have substantially strengthened the manuscript. The authors have demonstrated a commendable commitment to refining the clarity, depth, and methodological precision of their work. The study stands as a meaningful contribution to the field of cognitive aging and EEG-based biomarkers, offering insights into the potential resilience of older adults with high theta AP in cognitively demanding contexts. The expanded discussion on cognitive reserve, compensatory mechanisms, and broader biomarker integration enhances the interpretability and impact of the findings.

With these improvements, the manuscript is now well-positioned for publication, with only minor considerations for future research directions. The careful attention to prior critiques and the methodical approach in addressing them reflect the authors' commitment to high scientific standards.

Recommendation: Acceptance with minor considerations for future research directions.

**Do you want your identity to be public for this peer review?** For information about this choice, including consent withdrawal, please see our Privacy Policy

Reviewer #1: **Yes: ** Bochao Danae Lin

Reviewer #2: **Yes: ** Jacob A. Eder, PhD

---

## [Editor Report · Acceptance letter]

PONE-D-24-42123R1

PLOS ONE

Dear Dr. Fernández,

I'm pleased to inform you that your manuscript has been deemed suitable for publication in PLOS ONE. Congratulations! Your manuscript is now being handed over to our production team.

Kind regards,

on behalf of

Dr. Giulia Prete

Academic Editor

PLOS ONE